# ERP Indicators of Phonological Awareness Development in Children: A Systematic Review

**DOI:** 10.3390/brainsci13020290

**Published:** 2023-02-08

**Authors:** Katarina Stekić, Olivera Ilić, Vanja Ković, Andrej M. Savić

**Affiliations:** 1Laboratory for Neurocognition and Applied Cognition, Department of Psychology, Faculty of Philosophy, University of Belgrade, 11000 Belgrade, Serbia; 2School of Electrical Engineering, University of Belgrade, 11000 Belgrade, Serbia

**Keywords:** phonological awareness, phonological processing, ERP, dyslexia

## Abstract

Phonological awareness is the ability to correctly recognize and manipulate phonological structures. The role of phonological awareness in reading development has become evident in behavioral research showing that it is inherently tied to measures of phonological processing and reading ability. This has also been shown with ERP research that examined how phonological processing training can benefit reading skills. However, there have not been many attempts to systematically review how phonological awareness itself is developed neurocognitively. In the present review, we screened 224 papers and systematically reviewed 40 papers that have explored phonological awareness and phonological processing using ERP methodology with both typically developing and children with reading problems. This review highlights ERP components that can be used as neurocognitive predictors of early developmental dyslexia and reading disorders in young children. Additionally, we have presented how phonological processing is developed neurocognitively throughout childhood, as well as which phonological tasks can be used to predict the development of phonological awareness prior to developing reading skills. Neurocognitive measures of early phonological processing can serve as supplemental diagnostic sources to behavioral measures of reading abilities because they show different aspects of phonological sensitivity when compared to behavioral measures.

## 1. Introduction

Interest in the concept of phonological awareness rose steeply in the 1970s, alongside research in the development of reading abilities. At first, phonological awareness was defined as a single concept that refers to the awareness of sounds that make up the words we use in everyday language. The cumulative research on phonological awareness throughout the last six decades has shown that phonological awareness is rather an ability that consists of several different abilities, e.g., phoneme awareness and syllable awareness [1]. Some authors argue that there is a difference between phonological awareness in terms of sensitivity to sound similarities and in terms of segmental phoneme discrimination [2,3]. However, there are still authors opposing this view. Anthony and Francis [4] argue that “phonological awareness is a single, unified ability during the preschool and early elementary school years that manifests itself in different skills throughout a person’s development.” Nonetheless, most authors agree that phonological awareness is a skill that can be measured by assessing awareness of both smaller and larger language units that seem to differentially contribute to phonological awareness as a unified ability. Moreover, it is a skill that starts developing since the first interaction with letters and continues developing with further reading experience [5].

Research on phonological awareness has so far mainly focused on understanding predictors of children’s reading abilities; it has been widely related to children’s word reading skills across cultures, as well as reading disabilities [6,7,8,9]. The most common hypothesis in the literature with regard to developmental dyslexia is that dyslexia is a consequence of poor phonological awareness. This hypothesis is known as the linguistic hypothesis, postulated within the phonological deficit theory [10]. However, as Premeti, Bucci, and Isel [11] have shown in a recent review, an alternative hypothesis that proposes visual attentional deficits as the root of dyslexia also has empirical support. Another approach proposes the phonological mapping hypothesis, where reading disabilities are thought to emerge as a result of poor orthographic phonological mapping and an absence of left lateralization of neural responses to print sensitivity [12]. Thus, both phonological and visual deficits should be considered when discussing neural correlates of dyslexia. Despite this knowledge, phonological awareness is one of the abilities that has rarely been investigated neurocognitively with regard to developmental dyslexia.

To this day, there has been a vast amount of behavioral research analyzing the potential predictors of literacy and understanding the relationship between them. There have also been neurocognitive studies that have sought correlates of phonological awareness development and other reading predictors in the field of fMRI [13,14,15,16,17,18,19], but there have been far more attempts to find ERP correlates of reading development and reading disabilities (mainly due to EEG technology being more readily available to researchers). Thus, this study will further focus on ERP correlates, as they are also most likely to be used in reading disorder diagnostics.

One of the ERP components investigated with regard to early phonological processing is the rhyming effect (RE). The RE component is elicited by rhyme judgment tasks, which are typical N400 paradigm tasks. RE onsets 250–300 ms after the target and peaks at 400–450 ms, greatly overlapping with the N400 window. Coch, Grossi, Skendzel, and Neville [20] have shown that the RE onset is correlated with the level of phonological awareness. Where phonological awareness scores are higher, RE onsets on average 80 ms earlier, although there is no correlation with RE peak latency.

Wagensveld, Van Alphen, Segers, Hagoort, and Verhoeven [21] differentiate a reversed anterior RE and a typical N400 posterior RE, discussing that anterior and posterior locations may have distinct functions in the phonological processing of rhyme. An earlier N240 RE was also reported in studies, but this component seems to tap into different processes, such as phonological acoustic mismatch expectancy, which is only evident in word rhyming, but not in nonword rhyming [22,23].

Another difference between skilled readers and early readers lies in processing phonological/orthographic incongruence, where children find it more difficult to inhibit conflicting orthographic interactions. This cognitive effort elicits a broad N350 response in both hemispheres, which results in a reduced negative amplitude to conflict when compared to adults who have a more left-lateralized response. Inhibitory control [24] and shifting [25] have been shown to be predictive of phonemic awareness, thus the go/no go paradigm was used to explore the neural correlates of phonological processes. Kim et al. [26] showed that stronger reading skills predict an error positivity (Pe) response, but not error-related negativity (ERN). MMN, which reflects phonetic discrimination skills in the verbal domain, has also been shown to change even with fairly short reading interventions [27]. Dyslexics are shown to have a less pronounced MMN in response to auditory temporal and linguistic processing [28], as well as a low amplitude and long latency of P1 [29]. This hindered MMN response is thought to reflect impaired categorical perception of lexical tones [30].

Some earlier components have also been shown to be predictive of reading skills, such as the N170 response, which reflects experience with visual words. Reading expertise leads to left lateralization of the N170 response in both alphabetic and logographic scripts [31,32], driven by script familiarity. Children also have a different CNV response in rhyming tasks compared to adults, as the CNV amplitude lowers with age [20]. The CNV is more localized to fronto-temporal sites in children and more globally distributed across the scalp in adults.

Both fMRI and ERP research have shown that left lateralization is a strong predictor of reading abilities [32,33,34,35]. Dyslexic children have a more bilateral response to phonological material [36,37,38]. Transcranial stimulation has been shown to induce improvements in word reading efficiency in below-average readers [39] as well as phonological training [35,40]. Dyslexics have been shown to have atypical responses in both early and late phonological processing. The N170 effect in dyslexics shows a reduced level of print sensitivity, suggesting a deficit in visual orthographic processing during reading [11]. Children at risk of dyslexia also show no sensitivity to non-rhyming pairs in rhyme judgment tasks, which shows an absence of an analytical approach to sound similarity [41]. Moreover, differences in late components such as the N400 show that dyslexic children have problems with accessing phonological representations and capturing orthographic knowledge during reading [42].

### 1.1. Rationale

To date, there have been multiple narrative and meta-analytic reviews that tap into the role of phonological awareness in learning to read [43,44,45,46,47,48,49]. However, no reviews so far have focused on clarifying the neural correlates of phonological awareness. This review aims to explore ERP correlates of phonological awareness, as ERP methodology has most commonly been used to study this phenomenon neurocognitively. Additionally, our goal is to present a landscape of measures of phonological awareness and phonological processing that have been identified with the assistance of EEG methodology with regard to both normal and abnormal reading development.

### 1.2. Objectives

This study aims to summarize the various ERP measures identified in previous research to indicate the presence of phonological awareness (or absence thereof in the case of reading disorders). We aim to synthesize the findings of these studies to map ERP neural correlates of phonological awareness and early phonological processing in typically developing children and children at risk of or diagnosed with dyslexia, from infancy to their early teen years.

## 2. Method

### 2.1. Eligibility Criteria

During abstract screening, three reviewers adhered to the following criteria. (1) The study includes in the title or text one of the following keywords: phonological awareness, EEG, ERP, preschool, children, dyslexia; (2) the study reports original empirical data based on measures of phonological awareness and/or phonological processing; (3) the study uses a research design that includes EEG or ERP measures; (4) the study includes at least one sample of typically developing children, dyslexic children, poor readers, and children at risk of developing dyslexia or other reading disorders; and (5) the study is published in English.

### 2.2. Information Sources

The initial search was conducted by covering three databases (Scopus, PubMed, OpenAlex) and Google Scholar.

### 2.3. Search Strategy

The following keywords were used to obtain results from Google Scholar: *intitle: phonological awareness eeg or erp preschool children dyslexia*. Citations were excluded from the search. There were no limitations on the publication date. To search through Scopus, PubMed, and OpenAlex, the Publish or Perish software was used [50]. Since Publish or Perish does not recognize Google Scholar’s advanced search language, the following keywords were used to search through the three databases: *phonological awareness eeg erp preschool children dyslexia*. The search results of all databases were obtained in August 2022.

### 2.4. Selection Process

Three independent reviewers manually screened abstracts of 224 results uncovered by the initial search. After abstract screening and excluding studies that do not meet the eligibility criteria, each of the remaining studies was thoroughly read to look in more detail at ERP indicators of phonological awareness or phonological processing. During this phase, more articles were excluded if the text of the full study was not obtainable, or if it was uncovered that the publication does not fully fit the eligibility criteria.

Studies that were excluded during the abstract screening phase were those that do not fit at least one of the mentioned criteria. Unpublished manuscripts, preprints, and PhD dissertations were included in this review if they fit the eligibility criteria. Books, handbook chapters, and commentaries were excluded because they do not provide new empirical data. Conference proceedings were also excluded if not providing the full text of the article. All studies wherein it was not possible to determine from the abstract whether all of the criteria were fulfilled were moved to the phase of thorough reading, after which some were excluded.

Each step of the selection process is presented in Figure 1. Publications included in this review are presented in Table 1, Table 2 and Table 3, grouped by age. We identified three distinct age groups of interest for this review: (1) infants and toddlers (from birth to 4 years), preschoolers (from 4 to 7 years), and school-aged children (from 7 years to 15 years).

### 2.5. Data Collection Process

The list of results from Google Scholar and Publish or Perish database scraping were obtained in CSV format. Three reviewers went through both lists independently and analyzed the abstracts during the screening process. For the thorough reading phase, each reviewer obtained the results in PDF format, where it was possible to do so. During thorough reading, reviewers identified: (1) which method of ERP testing was used in the study; (2) what type of cognitive processing this method tests, and (3) which ERP neural correlates have been identified in the study to reflect these cognitive processes; and (4) what the final conclusions of the studies with regard to phonological awareness and/or phonological processing were.

### 2.6. Data Items

All studies that have utilized ERP or EEG methodology to neurocognitively measure phonological awareness and/or phonological processing were regarded as eligible for inclusion. The following variable data were obtained: (1) the name of the ERP component and (2) the time window (latency). We also collected the following data about the participants: (1) the number of typically developing participants (TD) and the number of dyslexic participants, or participants at risk of developmental dyslexia or other reading disorder, as well as poor readers (RD); (2) the age range (or age average); and (3) the native language of the participants. Finally, we identified which task or tasks were used to elicit the ERP components in the study.

### 2.7. Meta-Bias and Risk of Bias in Individual Studies

The Cochrane risk of bias tool was used to assess the risk of bias in individual studies [84]. We have assessed studies based on selection bias (whether there is a systematic baseline difference between groups) and performance bias (whether there is a systematic difference in the way groups are treated). On the selection bias criterion, out of the 40 studies, 2 were rated as being at high risk of bias, 31 were rated as low risk of bias, and 7 were rated as unclear risk of bias. On the performance bias criterion, none were rated as at high of risk of bias, 37 were rated as low risk of bias, and 3 were rated as unclear risk of bias. Results from the two studies [55,58] that were marked high risk should be interpreted with caution due to oversampling of dyslexic children and bilingual children, respectively. We have aimed to overcome this publication bias by including unpublished manuscripts, preprints, and PhD dissertations [85].

### 2.8. Confidence in Cumulative Evidence

The GRADE system was used to assess the quality of the results of the studies included in this review, and the key metrics that were taken into consideration were (1) the number of electrodes used to collect data; (2) the length and adequacy of the time window for the reported ERP component; (3) the baseline window; (4) the baseline correction methods; and (5) the filtering methods. If any of the aforementioned information was missing, the study was rated lower in quality. Lower grades were also given in cases wherein the methodological setup does not match the industry standard (e.g., if the N400 effect was recorded from an atypical location that does not include typical N400 electrodes, such as the Cz, Fz, or Pz). No studies were excluded at this stage of the review, as all studies were rated high (28) or moderate in quality (12).

### 2.9. Synthesis of Results

A narrative synthesis was employed to compile the results regarding the ERP components explored in relation to phonological awareness and phonological processes. The summary tables include essential extracted features from each of the studies (identified ERP component, time window, number of typically developing participants and their age, and the task that was used to elicit the ERP component in question). The results are discussed in the context of markers of phonological processing that could predict the development of phonological awareness in pre-reading children and beginner readers.

## 3. Results and Discussion

Out of the 40 papers reviewed here, 19 were focused solely on TD children, and 19 involved samples of both TD and children with a developed reading disorder or children at risk of developing reading problems (RD). Two studies focused only on RD children. In total, these studies interpreted results from 1720 children, out of which 1204 were TD children and 516 were children with RD. A total of 15 different languages were explored in these studies, with English (n = 17) and German (n = 8) being the most common, followed by Dutch (n = 5) and Finnish (n = 4).

### 3.1. ERP Markers of Phonological Processing in Infants and Toddlers

This review has shown that cognitive processes involved in phonological processing are different across development. When it comes to infants and toddlers (less than 4 years old), the predominant neuromarkers of phonological processing are receptive responses to sound stimuli.

MMR and ABR responses in particular have high predictive power to distinguish between children at familial risk of dyslexia and typically developing children [51,53]. Detecting temporal changes in tone patterns at 17 months predicts language development at 4 years, while ABR responses to clicks at 6 weeks predict the number of phrases understood as well as gestures and words produced by the age of 9 months. Aside from these early components, some late components have been identified in infants and toddlers as well [86,87]. Phonemic and auditory discrimination measured through nonword repetition elicited both early and late components that predict language development as well as writing skills.

#### 3.1.1. ABR

The typical ABR response in newborns consists of three subcomponents that are formed by three out of the five brainstem waves. The first (Wave I) mirrors the activity of the cochlear nerve; Wave II represents the activation of the cochlear nucleus, while Wave V represents activity at the lateral lemniscus. Out of the three, Wave V is most commonly used as an indicator of early auditory processing. In one study included in this review [53], Wave V latency measured in infants predicted language development in the span of three months. Infants who showed shorter Wave V latency in an ABR forward masking paradigm at 6 months of age had better language development scores at 9 months. This finding is in line with studies done in both adults and children that show response latency to phonological stimulation should typically decrease over time, while in troubled readers it remains increased [81]. Thus, ABR latency could be an early indicator of phonological skills, and a potential precursor of phonological awareness.

#### 3.1.2. MMN

Children that develop typically have reduced neural responses to phonological material when compared to troubled readers already in infancy in the early MMN time window. Faster responses to novel stimuli correlate with higher language performance [86]. Atypical MMN activation to speech sounds in infancy implies deficient development of phonological representations or connectivity to those representations in at-risk infants that later hinders access to the mental lexicon, and therefore also phonological awareness [52,74]. Children that are likely to develop reading problems have an atypical mismatch response (MMR), so these neural responses could be regarded as early indicators of phonological awareness. Aside from amplitude, response latency also predicts future reading problems. When it comes to at-risk children, there is some evidence that the MMR predicts future reading fluency, but not phonological awareness [51]. However, the authors note that phonological awareness was tested in second grade with children that have already started reading instructions, thus the MMR could still be predictive of early phonological awareness skills. This study also reports that the MMR is not elicited in toddlers at risk of developing dyslexia, indicating a difference in phonological discrimination skills already at 17 months. The authors conclude that temporal auditory processing differentiates young children at risk of dyslexia from controls, and thus could be a valuable marker for discovering dyslexic patterns early.

### 3.2. ERP Markers of Phonological Processing in Preschool Children

The second age group that was explored in this review is the preschool-aged group, i.e., children that have not yet started their formal reading lessons. Studies that investigated this group found a variety of both early and late components associated with phonological processing. The following paragraphs will focus on the most commonly reported components for the preschool age group.

The earliest components identified in preschool-aged children include an early MMN (ranging 100–200 ms), P1/P2 (ranging 90–215 ms), and N1 (ranging 180–290 ms). One study [61] reported an even earlier component P1’ that ranges between 6 ms and 88 ms. These early components are commonly elicited by target detection tasks (oddball paradigm), priming tasks, and visual differentiation tasks. The early components in preschoolers seem to tap into early phonological processes that distinguish language material from other types of stimulation. This auditory discrimination skill allows for phonemic analysis to develop, which is a direct measure of phonological awareness. The studies that identified early components showed that TD children have opposite early effects with word primes (N1/N200 reduction) vs. non-word primes (P1/N1 enhancement). Thus, with typical development in language comes a reduced mental effort for the processing of words. Phonemic analysis becomes easier for children as they learn to read, which is why early components that tap into discriminatory processes, such as the MMN, can serve as a good measure of phonological awareness development in preschool age as well.

#### 3.2.1. MMN

The latency, laterality, and amplitude of the MMN component seem to be indicative of different processes with regard to phonology in preschool-aged children. The laterality of the early and late MMN was significantly different in children with low versus typical rapid automatized naming ability, but this was not indicative of phonological awareness [55]. Although the MMN response is still elicited in preschool children, commonly with the auditory oddball paradigm, the latency changes with age. Compared to infants, preschoolers have a delayed MMN response that stretches to 600 ms [64], but it is still not adult-like. Children with specific language impairment (SLI) are impaired in discriminating both speech and non-speech information, eliciting no MMN response to either linguistic nor non-linguistic contrasts [56]. Similarly, children with hearing impairments show reduced amplitudes in the early parts of the late discriminative negativity component (LDN), which overlaps with the late MMN window. Thus, it seems that the MMN reflects a sensitivity to phonemic material that becomes more complex with age, shifting the latency to later windows.

However, the MMN is also task dependent, so if the stimuli presented are more complex, the latency will shift towards later time windows regardless of age to compensate for this complexity. Moreover, the group differences in MMN are so far thought to be speech-specific, since the auditory MMN is diminished or not elicited at all by pure tones in children with a phonological deficit [64]. With regard to amplitude, the late MMN amplitude is significantly greater in children with typical phonological awareness than those with low phonological awareness [55], indicating that children with typically developing phonological skills are more sensitive to subtle phonological contrasts. Late MMN amplitude could be a direct predictor of phonological awareness through indicating access to phonological representations [58,88].

The early MMN that peaks around 150–250 ms is thought to represent initial auditory change detection [58], while the late MMN that peaks in the range between 300 and 500 ms is thought to mirror attention switching between different forms of phonological material [55]. Other authors have suggested that the late MMN enables access to phonological representations, while an even later MMN response (450–600 ms) allows for phonological complexity processing [64]. There is also evidence that phonologically atypically developing children show a qualitatively different response to the same phonological stimuli when compared to their typically developing peers. These findings are interpreted in terms of phonological underspecification, wherein children with PD function at a developmentally less mature stage of phonological acquisition than their typically developing peers.

#### 3.2.2. N1

The N1 response is thought to reflect print sensitivity and script familiarity, while early N1 lateralization has been shown to be sensitive to language development in preschool children [57]. Stronger N1 activation has previously been reported with increasing literacy [89]. In order to learn to read, children need to master grapheme–phoneme correspondence, which requires developed phonological awareness.

The more pronounced left occipito-temporal negativity to words vs. nonwords is absent in non-reading kindergarten children [57]. Enhanced N1 left-lateralized amplitudes for phoneme matches compared to phoneme mismatches have been obtained for adults in the N100 time window, but not for non-reading children [59]. Therefore, the N1 component seems to develop with reading instruction, and becomes more lateralized as reading skills are developed. With regard to phonological awareness, this could mean that N1 lateralization is a key neural precursor in the visual domain for developing phonemic awareness and learning to read fluently.

#### 3.2.3. RE

With regard to late ERP components, studies in this review point out a unique late ERP response to phoneme priming in preschoolers [59]. Enhanced anterior positivity for phoneme mismatch has been obtained in both adults and children in the P350 time window. However, the effect is bilateral in pre-reading children, while it is left-lateralized already in beginner readers.

Another late component that emerges in pre-reading children is the RE, elicited by rhyme judgment tasks that typically include the instruction to decide whether words rhyme or not [90,91]. There is a noticeable correlation between success in this type of task and other phonological awareness tasks [20]. Those who have low correctness in rhyming tasks, typically fail in other, more complex phonological awareness tasks [33]. These findings led researchers to believe that rhyming is a stem ability that other phonological skills are built on. Children can typically detect rhyme well before they start learning to read.

The rhyming effect (RE) has been elicited by the rhyme judgment task in several studies included in this review. Wagensveld et al. [21] have shown that a distinct N450 RE can be elicited in second-grade children. However, kindergarten children who have not yet had reading instructions, do not show this effect, meaning that although they can detect rhyme, they do not seem to detect non-rhyming stimulation as easily. The second-grade RE is more global, distinguishing only rhyme processing, while in adults, this type of measurement in rhyming tasks typically distinguishes a fine gradient of phonological processing. The difference between early readers and nonreaders is also evident in the way they process rhyme; kindergarten children show an enhanced negative response to rhyming items, whereas adults and second-graders show an enhanced negative response to non-rhyming items, especially in nonwords.

As Wagensveld et al. [21] have proposed, reading instructions seem to be a necessary element in order for the RE to appear. No studies included in this review reported the RE component in children under the age of six [20,83]. However, as was already mentioned, the onset of RE, rather than its amplitude or latency, is predictive of phonological awareness skills [20]. Thus, sensitivity to non-rhyming rather than rhyming targets seems to be crucial for the development of phonological awareness.

#### 3.2.4. N400

The typical N400 response was also elicited in a few studies that resorted to the lexical decision task or orthographic/phonological congruency [61,77]. This component may be indicative of different levels of phonological awareness development as a function of N400 amplitude to incongruent stimuli, especially nonwords.

There is a significant developmental difference in the N400 response as this component matures. The N400 in adults is mostly semantic, while in preschoolers, it is phonological. Thus, the N400, in contrast to the N/P150 and N250, has been shown to predict various aspects of phonological development, being related only to phonological awareness contemporaneously, but predicting vocabulary across years [62]. This clearly represents a move towards a more semantic, adult-like function of the N400 over time.

In one MEG study, auditory and visual responses as well as audiovisual integration of letters and speech sounds were correlated with children’s behavioral cognitive skills [60]. The results revealed that auditory processing, especially the auditory processing in the late N400 time window, was the driving force for the correlation between sensory evoked fields and phonological skills. The authors of this study suggest that this 400 ms window could reflect the effect of speech–sound representations, as it is sensitive to phonological priming. Children with dyslexia have impairments in integrating phonological information into word-level representations, and this is visible in the N400 response [80]. Thus, it seems that the N400 phonological function in preschoolers serves as a precursor to developing semantic representations. The N400 may reflect the later stages of phonological awareness with regard to integrating sounds into meaningful units (i.e., words) after phoneme awareness has already started developing.

### 3.3. ERP Markers of Phonological Processing in School-Aged Children

The third group we analyzed were school-aged children that are beginning readers or have already significantly developed reading skills. What most of the studies that included a sample of children within this age group show is that the typical ERP responses of school-aged children are very similar to typical adult readers. We will highlight only the developmental differences in the following paragraphs, with more focus on the components that are novel in this developmental phase (i.e., P3, N2, and CNV).

Studies that included school-aged children also focused more on the developmental aspects of phonological processing in struggling readers, especially in the critical period in which reading should become fully automated. One such study explored how P300, N170, and N400 develop across time in struggling readers that had no reading interventions, emphasizing the non-significant correlation between behavioral and neural measures across reading development [82]. Namely, as struggling readers gain more experience with reading, they seem to develop better strategies to overcome their phonological deficits, resulting in better reading scores. However, neurocognitive markers of reading abilities do not follow this trend. This result shows that although struggling readers can improve their reading skills behaviorally, this comes at a higher cognitive cost to them because they have to extract more neural resources to achieve the same results as typical readers.

Other studies have shown that the auditory MMN response seems to mature early, as it remains an indicator of phonological complexity processing in both school-aged children and adults [67]. The MMN response at this age also shows that sensitivity to phonotactic probability remains impaired in dyslexic readers during school years [78]. The visual MMR as a response to mouth movements is more indicative of the distribution differences between dyslexic and typically developing school-aged children. Namely, children with developmental dyslexia show a more anterior MMR distribution, while typical readers show a posterior distribution [76]. Since the anterior MMR distribution is typically linked with auditory processing, it is suggested that dyslexic children might rely more heavily on the anticipation of the auditory codes to compensate for their phonological deficits.

When compared to monolinguals, typically developing bilinguals develop increased sensitivity to speech sounds, which is recognizable in the late negativity (LN) component [92]. When compared to their dyslexic peers, typically developing school-aged children present an increased degree of print sensitivity through the N170 component [68]. The N1 response has also been shown in typically developing preschoolers, while it continues to become more left-lateralized during school years, as reading becomes an essential part of the child’s everyday activities. The N400 and the late positive complex (LPC) are both attenuated in school-aged children with dyslexia, showing that these late cognitive processes are essential for grapheme–phoneme conversion and also for obtaining phonological access during school years. Moreover, the dyslexic N400 response at school age indicates that speech perception difficulties in dyslexia might have consequences for processing auditory words, as dyslexics have problems integrating phonological information into word-level representations [92].

#### 3.3.1. P3

One study included in this review that compared groups of children from 7 to 18 years of age [73] found that all groups elicit a P3 response, but there were differences in latencies. The distribution changed as a function of age and task. Latencies decrease with age (also shown in [81]), and the age differences remain significant by the age of 16. Although there is a large shift in cognitive effort for phonological tasks in the 9–10 age group, the difference is much greater for orthographic tasks. This type of processing difference across age groups indicates that phonological awareness is only one of the precursors of language abilities, while they continue to develop across teen years. Taylor [73] showed that both phonological and orthographic cognitive development happens during school years. Although orthographic tasks become much cognitively easier at the age of 9–10, phonological tasks seem to remain cognitively demanding for older children, especially dyslexic children who do not show this developmental pattern, as their P3 latency does not decrease with age [81]. P3 latencies are significantly longer in dyslexic than control children in both phonological and semantic tasks.

Coch et al. [22] showed that the P300 can be elicited by a rhyme judgment task, particularly to rhyming targets, while non-rhyming targets had a later onset. This finding is in line with the memory-updating hypothesis that proposes nonmatches require more updating resources than matches. When it comes to rhyming tasks, the RE and P300 overlap, but show two distinct effects with regard to rhyme processing; the RE is more specific to phonological ability, while the P300 represents a more general updating ability.

Neither of the studies included in this review showed shifting lateralization of the P3 component with age. P3 is thought to represent memory updating, which is required for processing phonological information but is more bilateral in nature. However, one study included in this review showed significant anterior-posterior distributional group differences in the P3 between dyslexics and normal readers. Dyslexics show frontal negativity and a more posteriorly located P3 compared to normal readers [81].

The P300 latency is also indicative of differences between dyslexic and typical readers when their speed of processing visual-orthographic and auditory-phonological information is compared [79]. In normal readers, there is a gap between the two processing pathways, the visual-orthographic path being typically faster. Dyslexic readers do not only process both visual and auditory linguistic information more slowly than typical readers; their auditory-phonological pathway is much slower than their visual-orthographic pathway when compared to this processing gap in normal readers. Moreover, in one case mentioned in this study, the processing gap was inverted, i.e., the child processed auditory information more quickly than visual information. These findings show that the P300 can be useful not only for distinguishing dyslexics from typical readers but potentially for uncovering subtypes of dyslexia and comorbidities.

#### 3.3.2. N2

Only one study mentioned the N2 in preschoolers as a potential indicator of early phonological awareness, thought to represent an explicit learning effort with regard to attention control mechanisms [61]; meanwhile, in school-aged children, the N2 seems to be a more relevant indicator of phonological skills. The N2 is already adult-like by the age of nine [93], which is consistent with the hypothesis that the acceleration of spoken-word processing continues beyond age seven, and that the negative components that tap into phonological learning mechanisms become more specialized for semantic processing with age.

N2 latency, which is related to stimulus evaluation and categorization, decreases with age. Differences between sequential age groups become insignificant after the age of 11 [62]. This shows that the N2 component does develop up to a certain point in the child’s phonological development, but it becomes adult-like as soon as the child masters reading skills. However, no lateralization with regard to N2 development has been shown, indicating that the N2 reflects processes that are not phonological in nature, but are required for the development of phonological awareness, similar to P3. Thus, the N2 in school-aged children may serve as an indicator of fully developed phonological awareness for the native language, and potentially for the second language [94], similar to N400 with preschoolers.

#### 3.3.3. CNV

One component that appears in school-aged children and adults, but not in younger children, is the contingent negative variation (CNV), which is thought to index anticipation, expectation, or motivation in preparing for a response, but also short-term memory. The CNV matures with age, as reading becomes automatized, resulting in a decreased amplitude in adults. Subjects with higher reading and spelling scores have a larger anterior CNV left asymmetry [66].

However, a few studies included in this review did not confirm that the CNV component reflects phonological short-term memory [20,22,66]. There was no correlation between the CNV response and short-term memory tasks, such as digit span or rapid automatized naming. The CNV also does not correlate with RE [66]. These results suggest that the CNV might not be phonologically specific, rather indicating a general readiness to respond. Thus, the interpretation of the CNV in the context of phonological processing requires further research.

## 4. General Discussion

So far, we have reviewed articles that have measured different ERP components that are related to phonological awareness, either empirically or theoretically. Some of these components have been related more closely to phonological awareness than others, and those are the MMN, N1/N170, RE/N400, and P3. These components stood out as more solid indicators of phonological awareness at various stages of development. However, some inconsistencies between the studies were noticed. Several studies included in this review showed that there is a significant correlation between behavioral measures of phonological awareness and the MMN response [55,58,65,71] while other studies failed to show this correlation [51,66,78]. There are also studies that have shown a positive correlation between MMN and phonological awareness exists only in children with developmental dyslexia, but not in typically developing readers [76], suggesting that the MMN could still be a useful neuromarker of reading disabilities. Where there is a correlation, the late MMN amplitude is typically greater in children with typical phonological awareness ability than those with low ability. However, since the MMN component matures early with regard to phonological skills, it loses most of its predictive value with age, and thus could serve as an indicator of phonological awareness development only in preschool children.

The N1 and N170 components are discussed with regard to phonological awareness where sensitivity to print is involved, while development of reading is heavily dependent on the lateralization of these components. Studies with blind children have shown that development of phonological awareness is hindered without visual input [95,96], suggesting that the nature of orthography also affects phonological awareness. One study included in this review showed that phonological awareness predicts left lateralization of the visual components [69] while the importance of orthographic training for dyslexic children is also emphasized [68]. However, not all studies showed that visual components are related to phonological awareness development [60]. The N1 is not left lateralized in pre-reading children, which is in line with the phonological mapping hypothesis [69], but starts lateralizing with early reading instructions [12]. The causal effects are not yet clear, i.e., whether lateralization is a result of phonological awareness development, or a precondition. Thus, further research is required to show whether visual components can become predictive of reading before formal reading instructions, but with phonological awareness development. Further research could also explore the changes in N1/N170 lateralization and amplitude prior to and after phonological awareness training. As far as current knowledge stretches, visual components can be used to predict phonological awareness in children no younger than school age.

The P3 latency has been directly related to phonological awareness in several studies included in this review, but does not seem to be predictive of phonological awareness specifically. The most reliable neural predictor of phonological awareness seems to be the rhyming effect (RE), which overlaps with the N400 time window, so some studies have found the typical N400 effect with the rhyme judgment task as well (e.g., [41]). The neural response in rhyme judgment tasks was shown in multiple studies included in this review to highly correlate with future reading skills. Rhyming is a rudimentary phonological awareness skill that children develop implicitly before formal reading instructions. The RE matures in middle childhood and does not show differences across ages, but dyslexics have been shown to have a problem with rhyming, even as late as adolescence with visual and auditory rhyming [70,77,97]. Thus, the rhyme judgment paradigm could be used as a task to identify early indicators of potential reading disorders. Children who are less sensitive to rhyme incongruity develop phonological awareness later and also have more issues with reading. They also show an aberrant or atypical response to rhyme stimuli, barely differentiating rhyming from non-rhyming targets [11,41], or having a more pronounced N400 response to rhyme targets. However, not all studies showed a correlation between the RE and the phonological awareness tasks in typical readers [66,83]. Where there is a correlation, the latency of the RE (the onset in particular), rather than the amplitude, is related to phonological awareness.

When different languages are considered, the depth of orthography may play a significant role in the discrepancy of some results. As Coch [82] pointed out, some results established in Italian and French are not replicated in English-speaking children. On the other hand, studies included in this review that were obtained on more shallow orthographies (e.g., Italian, Finish, German, and Dutch) mostly showed that phonological awareness is correlated with ERP measures of reading development. Wehner [74] points out that phonemic awareness deficits do not persist into adulthood in shallow orthographies such as Dutch, but do persist in deeper orthographies such as English. Thus, ERP measures may be more indicative of early phonological awareness deficits in both shallow and deep orthographies, while they could still serve as diagnostic measures of reading impairments in deep orthographies later in life.

With regard to the tasks used in the studies included in this review, it seems that the correlation between ERP measures and phonological awareness tasks is modulated by the nature of the task itself. If the task is more dependent on working memory, e.g., the phoneme deletion task, it is less likely to correlate with the N400 [78] and MMN amplitudes, but more likely to correlate with ERP indicators of working memory such as the P3 [36,73]. Since tasks that measure phonological awareness are not only phonological in nature, and neither is reading, further research could examine a mixed approach to dyslexia diagnostics, where correlations between specific ERP components and phonological awareness measures can serve as indicators of potential reading problems.

## 5. Conclusions

The present review is, to our knowledge, the first systematic review that aims to explore and summarize neural correlates of phonological awareness. We have shown that neural indicators of phonological awareness are both age and task dependent, while there are effective measures of phonological processing for indicating potential reading problems as early as infancy. Moreover, this review has tried to provide theoretical assumptions that tie together different phonological tasks and measures in order to show that phonological processing does indeed develop throughout childhood, even after children develop phonological awareness and begin to read. From the perspective of the linguistic hypothesis, these findings may be relevant for future research exploring early signs of developmental dyslexia and other reading problems.

We have reviewed a wide range of ERP components that were studied with regard to early and late phonological processing with the aim of identifying predictors and indicators of phonological awareness, showing several ways in which phonological awareness can be measured indirectly through a variety of tasks such as rhyme judgment, target identification, lexical decision, and priming tasks. Since the studies included in this review show no clear correlation between neural and behavioral measures for establishing developmental changes in phonological processing, this work also highlights the need for including neurocognitive methods when diagnosing reading problems, which is also a practical implication of this review. Neurocognitive measures provide more insight into the ways that struggling readers process phonological information. Thus, further studies could focus on developing ERP protocols that could directly test the link between phonological awareness and readiness to read, both as an indicator of normal and abnormal reading development.

## 6. Limitations

This study was conducted in order to explore the ways that phonological awareness can be studied using ERP methodology. However, no direct measures of phonological awareness were identified in this review. Moreover, currently available phonological awareness tasks may not even tap into phonological processing aspects of the ongoing word recognition process due to their complexity and difficulty [77]. Such findings show that there is a need to develop a novel ERP task that could directly tap into phonological awareness and show whether a child is capable of phonemic segmentation and analysis, even if the behavioral results show the opposite. 

The pool of studies included in this review consisted solely of studies that directly referred to phonological awareness with regard to ERP results in the context of reading development and/or developmental dyslexia and other reading disorders. Studies that investigated the same components in this context but did not relate the results to phonological awareness were not included in this review. This does not allow us to conclude that the list of components identified in this review is finite, and neither does it mean that these components are reliable direct measures of phonological awareness. This review should serve mainly as an overview of the currently available literature that connects ERP findings with phonological awareness. Moreover, since not many longitudinal studies were included, it is difficult to say at which point in the child’s development other auditory and visual skills are developed enough to support phonological awareness, and whether ERPs measured at two points in time (e.g., at infancy and as the child begins reading instructions) are sufficient to draw a solid conclusion about how phonological awareness is neurally developed. The early ERP measures reflecting auditory and visual abilities may present neural foundations for the development of phonological awareness and be indicative of its development, but may not reflect the distinct neural measure of phonological awareness itself.

Another limitation of this review is that it was predominantly based on studies that included children whose native language was English. The systematic review approach we have taken for this review prevented hand-picking articles that would potentially fit the search criteria. Future review studies could allow hand-picking in order to involve more references that are underrepresented by the search filters.

This study was conducted with the aim of providing an overview of the ERP components studied in relation to phonological awareness by summarizing and synthesizing previous findings. In order to provide a more in-depth understanding of the applicability of these ERP measures to directly tap into phonological awareness, further research is required.

## Figures and Tables

**Figure 1 brainsci-13-00290-f001:**
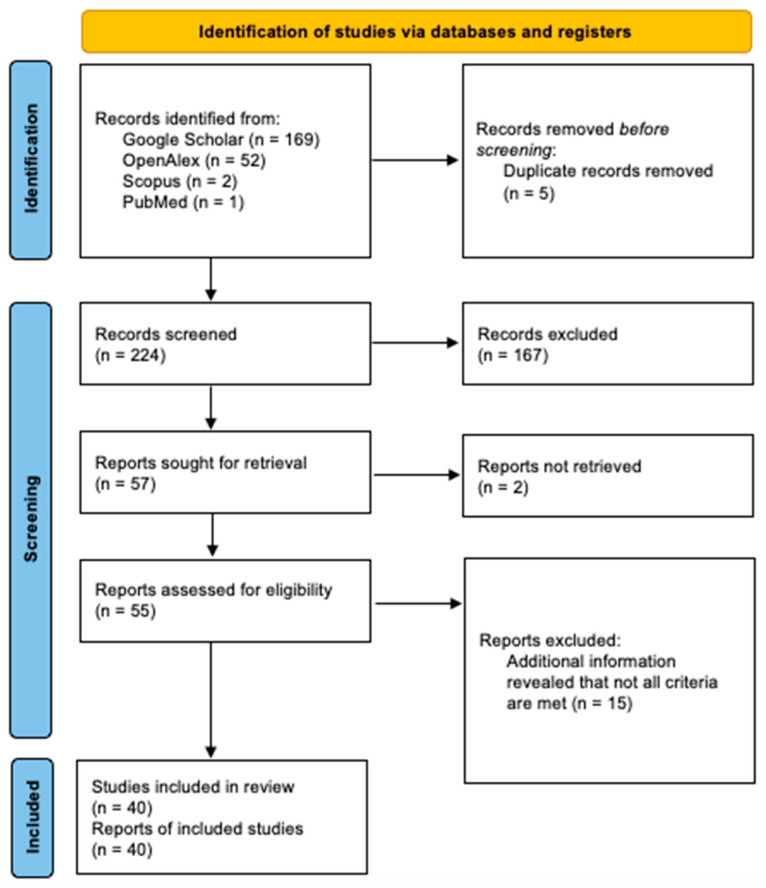
PRISMA diagram of the study selection process.

**Table 1 brainsci-13-00290-t001:** Overview of the publications selected for review; studies with infants and toddlers.

Study	ERP Components	Time Window (ms) ^1^	Age	TD (RD)	Task	Language ^2^
Van Zuijen et al. [51]	MMR	260 ± 30	17 mo	12 (12)	Auditory oddball	Dutch
Lohvansuu et al. [52]	Undefined	Late 370–470	6 mo	22 (26)	Auditory oddball	Finnish
Chonchaiya et al. [53]	ABR (Wave V)	0–74.67	6 w 9 mo	71	Hearing screening forward masking	Chinese, English
Guttorm et al. [54]	Undefined	Late 540–630	38.3–41.7 w	20 (23)	Auditory stimulation during sleep	Finnish

Note ^1^ refers to the time window where the peak of the effect was reported (post stimulus onset); ^2^ language of the participants included in the study; TD—typically developing; RD—reading disorder; mo—months; y—years; w—weeks.

**Table 2 brainsci-13-00290-t002:** Overview of the publications selected for review; studies with preschool children.

Study	ERP Components	Time Window (ms)	Age	TD (RD)	Task	Language
Norton et al. [55]	MMN	Early 100–200 Late 300–500	4.10–6.8 y	65 (101)	Auditory oddball	English
Davids [56]	MMN	80–1502	5–6.5 y	25 (25)	Auditory oddball	Dutch
Wagensveld et al. [21] preschoolers	N450 (RE)	250–400	5.7 ± 0.4 y	22	Auditory rhyme judgment	Dutch
Brem et al. [57]	N1	188–289	5.7–6.9 y	13 (12) + 10 (11) ^1^	Modality judgment	German
Linnavalli et al. [58]	MMN LDN	200–250 (225–275) 375–425	5–6 y	57 (13)	Phoneme processing	Finnish, Russian, Estonian, Albanian, Somali, Swedish, English
Schild et al. [59] preschoolers	N100 P350	100–300 300–400	5.8–6.9 y	24	Stress priming Phoneme priming	German
Xu et al. [60]	N1m N2m LC (N4) P1m N170m	amp. 104 ms amp. 243 ms amp. 413 ms amp. 100 ms amp. 209 ms	6–11 y	29	Auditory and visual oddball	Finnish
Studer-Eichenberger et al. [61]	P1′ P1 P2 N2 N4 LDN	6–88 100–116 188–214 232–282 370–450 462–528	4.2–7.5 y	13 (13) ^2^	Multi-feature paradigm	German
Stites & Laszlo [62]	N/P150 (N170) N250 N400	101–240 211–280 251–400	6–13 y	44	Name detection	English
Molfese et al. [63]	Factor 1 Factor 2 Factor 3 Factor 4	424–700 272–560 152–376 0–176	42–54 mo	33	Discrimination identification	English
Bitz et al. [64]	MMN	300–450 450–600	6.1–7.9 y	15 (19)	Auditory oddball	German
Coch et al. [20]	RE	360–420 360–420 480–540	6 y 7 y 8 y	19 18 17	Rhyme judgment	English
Schaadt et al. [65]	MMN	250–400 300–450 350–550 450–800 600–800 500–900	4.4–6 y	14 (15)	Auditory oddball	German

Note ^1^ intervention groups + control groups; ^2^ hearing impairment.

**Table 3 brainsci-13-00290-t003:** Overview of the publications selected for review; studies with school aged children and beginner readers.

Study	ERP Components	Time Window (ms)	Age	TD (RD)	Task	Language
Grossi et al. [66]	N120 P120 N180 N120 P200 N350 N400p RE CNV	75–200 50–175 150–250 175–200 160–300 200–500 300–600 250–600 600–1167	7–8 y 9–10 y 11–12 y 13–14 y	10 16 15 20	Visual rhyming	English
David et al. [67]	MMN LDN	180–250 380–520	6.6–10.7 y	22	Auditory oddball	French
Hasko et al. [68]	N170 N400 LPC	170–290 330–460 600–900	8.15 ± 0.27 y	29 (52)	Phonological lexical decision	German
Sacchi & Laszlo [69]	N170	160–190	10–12 y	20	Visual differentiation	English
Sun et al. [36]	P3a ^1^ MMN ^2^	233–293 ^1^ 273–333 ^1^ 110–170 ^2^ 116–176 ^2^	9–11 y	47	Passive oddball	Chinese (Mandarin)
Lovrich et al. [70]	N480 P800	undefined	11.64 y	10 (9)	Rhyming classification Semantic classification	English
Männel et al. [71]	MMN CPS	250–450 1000–1500	11 y	22 (22)	Passive listening Vowel length discrimination	German
Weber-Fox [72]	N350	200–500	9–10 y	20	Visual rhyme judgment	English
Taylor [73]	N2 P3	542 ± 46 475 ± 70 414 ± 48 434 ± 40 374 ± 61 766 ± 56 682 ± 51 638 ± 53 649 ± 49 587 ± 25	7–8 y 9–10 y 11–12 y 12–13 y 14–15 y	84	Visual–to–auditory translation	English, French
Wehner [74]	MNE	140–300 200–300	7–13 y	15 (15)	Attended oddball task Auditory discrimination task, Auditory sentence task	English
Malins et al. [75]	PMN N400 late N400	260–320 350–500 500–600	10.5 y	17	Picture–word matching	Chinese (Mandarin)
Varga et al. [12]	N1	164–282 165–302	7.08–9.29 y	41	Same–different paradigm	Hungarian
Coch et al. [22]	P50 N120 P200 N240 early CNV late CNV N400 P300	12–100 50–200 150–250 200–300 300–800 800–1200 200–700 400–1500	7–8 y 9–10 y 11–12 y	15 14 13	Rhyme judgment	English
Schaadt et al. [76]	vMMR	300–450 450–600 600–750 750–900 900–1050	9.57–9.62 y	34 (33)	Visual oddball	German
Bonte & Blomert [77]	P1 central N1 lateral N1 N2 N400a N400b	80–120 120–170 140–240 250–350 400–600 600–800	7–10 y	8 (10)	Auditory lexical decision task	Dutch
Noordenbos et al. [41]	N400	454–696	6–7.75 y	29 (30)	Rhyme judgment	Dutch
Spironelli et al. [35]	N150	170–190	118.64 ± 21.56 mo	(14)	Linguistic matching Phonological matching Orthographic matching	Italian
Bonte et al. [78]	MMN	360–580	8.2–9.5 y	14 (12)	Passive acoustic oddball	
Breznitz [79]	P200 P300	179.4–299.1 315.1–411.6	9.5–10.9 y	20 (20)	Linguistic and nonlinguistic auditory and visual tasks Homophone–homograph decision Rhyme decision	Hebrew
Desroches et al. [80]	N280 N400	250–330 430–680	8–11 y	15 (14)	Picture–word matching	English
Taylor & Keenan [81]	N2 P3	352–535 566–777	7–12 y	25 (25)	Semantic task Phonological task	English
Coch [82]	N170 N400 P300	150–300 300–500 400–700	110.9 ± 7.6 mo	(17)	Lexical task Phonological task Memory task	English
Wagensveld et al. [21] second graders	N450 (RE)	200–450	7.6 ± 0.6 y	20	Auditory rhyme judgment	Dutch
Schild et al. [59] beginner readers	N100 P350 extended processing	100–300 300–400 400–1000	7.2–8.11 y	23	Stress priming Phoneme priming	German
Coch et al. [83]	N450 (RE)	350–550	6.0–8.11 y	16	Visual rhyme judgment	English

Note ^1^ refers to the P3a time window in the study by Sun et al.; ^2^ refers to the MMN time window in the study by Sun et al.

## Data Availability

The full list of studies included in this review can be found at https://docs.google.com/document/d/1_RYdN_l9GnskrJtTF1it1_lpO6qRiY_3LRNQ5UXb2WY/edit, last accessed on 6 February 2023.

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
