# Peer review of "ERP Indicators of Phonological Awareness Development in Children: A Systematic Review"

_brainsci, 2023, doi:10.3390/brainsci13020290_

Round 1
Reviewer 1 Report
The review by Stekić et al. addressed the topic of neural correlates of phonological awareness in developmental population. The topic is of potential interests, since – to the best of my knowledge – there is not any systematic review on these specific aspects.
Despite the above, I have some serious concerns with the criteria for selecting articles: Including “in title or text either of the keywords: phonological awareness, EEG, ERP” does not imply that the ERP paradigm taps phonological awareness. In fact, most of the infants/toddlers studies (but also some of the children studies) used ERP paradigms that tapped basic auditory processing skills (such as ABR) or basic speech processing (such as oddball auditory paradigm with syllables). The fact that phonological awareness was separately assessed behaviourally does not allowed the authors to include the ERP components among those tapping phonological awareness. Much more studies have used the same tasks of basic auditory processing or speech processing in typical and atypical developmental populations but were not included in the review. I suggest to repeat the article selection and only include those articles using ERP paradigm tapping phonemic awareness.
A note a caution also concerns the inclusion of bilingual populations.
I do not agree with the statement that the standard for recording N400 is from central and frontal electrodes and not from parietal electrodes (page 4, lines 178-180). I do not think that it is a reason to exclude studies. Please report exactly how many studies were rejected at this stages and exactly the reason for doing so.
The flow of the introduction is unclear. For example, I suggest to anticipate the interests for the association between phonological awareness and reading development in both typical and atypical populations (i.e. dyslexia).
Children between 0 and 4 years of age should not be referred to as “infants”, please use infants and toddlers instead
Please correct typo at line 363
Author Response
The responses to Reviewer 1 have been provided in the attached Word file.

Reviewer 2 Report
This article provides a systematic review on phonological awareness development in children. While it identified the ERP components related to phonological awareness and phonological processing, the protocol needs to follow PRISMA guidelines closely with an in-depth analysis and discussion. A major revision is recommended.
1. According to the authors, the definition of phonological awareness refers to "a skill that starts developing since the first interaction with letters and continues developing with further reading experience." Even though phonological awareness is known to be associated with reading, its early development (for example, ability to discriminate phonemes) does not depend on reading. The review article needs to frame the scope of the review more clearly. Is it more focused on the ERP indices of normal development or on the ERP markers/precursors of dyslexia? If it is the latter, please consider revising the title.
2. The review needs to provide more in-depth analyses than what a reader can get from going through the highlights, abstracts and conclusions of the selected articles. There are a number of important issues that should be examined.
a. Please follow standard PRISAM protocol in searching and selecting articles and performing systematic review (See a recent tutorial at https://doi.org/10.1044/2022_JSLHR-21-00607) although meta-analysis may not be applicable.
b. What languages are represented in the selected articles? What selection criteria were used? What is consistent or inconsistent across studies in their findings? Are there language-universal and language-specific patterns in the findings? Why are studies on East Asian languages such as Chinese excluded? Why are some ERP studies on the precursors of dyslexia excluded and yet an ABR study was included? Here I provide some examples (Please note this is not an exhaustive list.)
Meng, X., Sai, X., Wang, C., Wang, J., Sha, S., & Zhou, X. (2005). Auditory and speech processing and reading development in Chinese school children: Behavioural and ERP evidence. Dyslexia, 11(4), 292-310.
Meng, Z. L., Liu, M. L., & Bi, H. Y. (2022). Spatial and temporal processing difficulties in Chinese children with developmental dyslexia: An ERP study. Dyslexia, 28(4), 416-430.
Molfese, D. L., Molfese, V. J., Key, S., Modglin, A., Kelley, S., & Terrell, S. (2002). Reading and cognitive abilities: Longitudinal studies of brain and behavior changes in young children. Annals of Dyslexia, 52(1), 99-119.
Moll, K., Hasko, S., Groth, K., Bartling, J., & Schulte-Körne, G. (2016). Sound processing deficits in children with developmental dyslexia: An ERP study. Clinical Neurophysiology, 127(4), 1989-2000.
Lallier, M., Tainturier, M. J., Dering, B., Donnadieu, S., Valdois, S., & Thierry, G. (2010). Behavioral and ERP evidence for amodal sluggish attentional shifting in developmental dyslexia. Neuropsychologia, 48(14), 4125-4135.
Zhang, Y., Zhang, L., Shu, H. Xi, J., Zhang, Y., & Li, P. (2012). Universality of categorical perception deficit in developmental dyslexia: An investigation of Mandarin Chinese tones. Journal of Child Psychology and Psychiatry, 53, 874–882.
c. If no regression or correlational analysis between the ERP components and behavioral measures of phonological awareness or phonological processing was performed, it seems unclear whether the ERP markers can reliably reflect or predict reading ability and language skills.
d. The main parts of the review focus on the different ERP components in pre-school children and school-aged children. It is my impression that each part needs to have more in-depth analysis across studies rather than just stating Components A, B, C, etc. are all potential ERP markers. If some component such as CNV or behavioral measure was not tested in younger children, it is most likely that this measure is age-dependent and there is not a readily available protocol for testing younger children.
3. What are the limitations of this review? What theoretical and practical implications does it provide?
Author Response
The responses to Reviewer 2 are provided in the attached Word document.

Round 2
Reviewer 2 Report
Thank you for providing the detailed responses and major revision. One remaining concern is about a conceptual loophole that cannot be easily addressed with the PRISMA protocol for systematic review. Please note that standardized tests of phonological awareness (PA) typically include behavioral subtests on rhyming, segmentation, sound isolation, deletion, substitution, blending, phoneme-grapheme correspondence, and phonemic decoding. In this regard, none of the ERP components and associated experimental protocols were originally intended to address phonological awareness development. Just because researchers in a study (for example, MMN study) discussed their results in relation to phonological awareness does not mean that the MMN itself is or is not a measure of phonological awareness. Conversely, many ERP (MMN in particular) studies that examined phonetic and phonological processing in early infancy and childhood were excluded in the systematic review because the researchers of those studies did not specifically tie their findings with phonological awareness, but this does not mean those ERP results have nothing to do with phonological awareness. Failing to establish direct statistically significant correlations between ERP components and phonological awareness test scores also does not mean the ERP components have nothing to do with phonological awareness.
Another concern is that longitudinal studies using early ERP measures in infancy and childhood and later language measures are required to address how the early ERP measures may predict later phonological awareness and language skills (including reading). But research grants typically have a limited time span (say 3 years), which may be insufficient for the longitudinal design as reading acquisition is a fairly long process during schooling. The conceptual loophole also applies to the longitudinal approach in the sense that the ERP components measured in infants and toddlers may reflect general neural sensitivity to auditory or phonetic cues in speech sounds. They may be considered auditory foundations for phonological awareness but not necessarily PA per se.
These points could be incorporated in the discussion section as limitations of the study.
Author Response
Thank you for an additional round of comments. These are all valid points that were mentioned, and we have now summarized them in the Limitations section of the paper.